# Association of multi-phase rates of force development during an isometric leg press with vertical jump performances

**Kodayu Zushi**[1¤]*, **Yasushi Kariyama**[2], **Ryu Nagahara**[3], **Takuya Yoshida**[4], **Amane Zushi**[5], **Keigo Ohyama-Byun**[4], **Mitsugi Ogata**[4]

1 Graduate School of Comprehensive Human Sciences, University of Tsukuba, Tsukuba, Ibaraki, Japan, 2 National Institute of Fitness and Sports in Kanoya, Kanoya, Kagoshima, Japan, 3 Japan High Performance Center, Nishi-ku, Tokyo, Japan, 4 Faculty of Health and Sport Sciences, University of Tsukuba, Tsukuba, Ibaraki, Japan, 5 Faculty of Sport Science, Yamanashi Gakuin University, Kohu, Yamanashi, Japan

¤ Current address: Faculty of Economics, Shiga University, Hikone, Shiga, Japan
* kodayu-zushi@biwako.shiga-u.ac.jp

**Data Availability Statement:** All relevant data are within the paper and its Supporting information files.

## Abstract

### Purpose

This study aimed to elucidate characteristics of explosive force-production capabilities represented by multi-phase rate of force developments (IRFDs) during isometric single-leg press (ISLP) through investigating relationships with countermovement (CMJ) and rebound continuous jump (RJ) performances.

### Methods

Two-hundred-and-thirty male athletes performed ISLP, CMJ with an arm swing (CMJAS), and RJ with an arm swing (RJAS). IRFDs were measured during ISLP using a custom-built dynamometer, while CMJAS and RJAS were measured on force platforms. The IRFDs were obtained as rates of increase in force across 50 ms in the interval from the onset to 250 ms. Jump height (JH) was obtained from CMJAS, while RJAS provided JH, contact time (CT), and reactive strength index (RSI) values.

### Results

All IRFDs were correlated with CMJAS-JH ($\rho$ = 0.20–0.45, $p \leq 0.003$), RJAS-JH ($\rho$ = 0.22–0.46, $p \leq 0.001$), RJAS-RSI ($\rho$ = 0.29–0.48, $p < 0.001$) and RJAS-CT ($\rho$ = −0.29 to −0.25, $p \leq 0.025$). When an influence of peak force was considered using partial rank correlation analysis, IRFDs during onset to 150 ms were correlated with CMJAS-JH ($\rho_{xy/z}$ = 0.19–0.36, $p \leq 0.004$), IRFDs during onset to 100 ms were correlated with RJAS-JH and RJAS-RSI ($\rho_{xy/z}$ = 0.33–0.36, $p < 0.001$), and IRFD during onset to 50 ms was only correlated with RJAS-CT ($\rho_{xy/z}$ = −0.23, $p < 0.001$).

**Funding:** This research was supported by the Japan Society for the Promotion of Science (JSPS) Grant-in-Aid for Scientific Research (grant number 22K21206) awarded to KZ.

## Conclusion

The early phase (onset to 150 ms) IRFDs measured using ISLP enabled the assessment of multiple aspects of leg-extension strength characteristics that differ from maximal strength; these insights might be useful in the assessment of the athletes' leg-extension strength capabilities.

## Introduction

Evaluation of force-production capability is important to monitor athletes' preparation and training adaptation level such that they can train effectively. In actual dynamic exercises (e.g., sprinting, jumping, cycling, change of direction), leg-extension force-production capability is an essential factor for achieving better performances. During most dynamic movements, ground reaction forces are generated in approximately less than 300 ms [1]. Therefore, the ability of leg extensors to generate explosive forces is integral to better performance in various sports [2]. The explosive leg-extension force-production capability has been evaluated via rate of force development (RFD) determined using a force-time curve during an explosive voluntary contraction test [3].

The RFD is measured during isometric force production (IRFD) to minimize injury risk and influence athletes' technical differences [3]. The IRFDs measured from the onset to 100 ms during an explosive knee extension test are primarily influenced by rapid neural activation in agonist muscles and the contractile capacity of a muscle-tendon unit, while the corresponding IRFDs are not strongly associated with peak force (PF) during maximal voluntary contraction [4, 5]. In contrast, the relationship between IRFDs and the PF during the same test increases gradually after 100 ms from the onset [4, 5]. Therefore, measuring multi-phase IRFDs from the onset to the PF enables the assessment of multiple aspects of force-production capabilities in athletes. To measure the IRFDs, an isometric mid-thigh pull (IMTP) or squat (ISq), which is similar to actual sporting movements in terms of trunk and leg postures, has been used [6]. However, IRFDs measured using IMTP and ISq are partially affected by the strength of arms and/or trunk muscles, which is a limitation of IMTP and ISq. As another IRFD measurement modality, an isometric leg press (ILP) can produce higher PF than that by IMTP and ISq [7, 8] and can be performed without any influence of arm and/or trunk muscle strength. In addition, injury risks and unnecessary arm and trunk muscle fatigues can be avoided during the IRFD measurement using ILP.

To monitor an athlete's specific force-production capability using IRFDs, it is necessary to understand the capabilities measured using IRFDs by investigating the relationships between IRFDs and dynamic exercise performances. Thus, previous studies have investigated relationships between IRFDs during IMTP or ISq and sprinting, jumping, agility, and cycling performances [6, 9]. The ILP is measured in a posture different from that for IMTP and ISq (ILP in a seated position versus IMTP and ISq in a standing position). This postural difference results in differing geometric arrangements of the body segments (shanks, thighs, and trunk) in relation to the force vector between the test modalities [7, 8]. Thus, the relationships between IRFD measured using ILP and dynamic exercise performances might differ from the corresponding relationships using IMTP or ISq. However, to our knowledge, only a few studies have investigated the association between IRFDs during ILP and dynamic exercise performances [10, 11]. In a study by Marcora and Miller [10], including fourteen active male postgraduate students who had experienced resistance training within at least 6 months, IRFD during the ILP was

correlated with countermovement jump height (CMJ-JH). Moreover, in a study by Häkkinen and Keskinen [11], including six sprinters and jumpers, seven powerlifters, and seven endurance swimmers, IRFD measured using ILP was higher in sprint athletes than in powerlifters and endurance swimmers. As the studies contained small sample sizes, the results may have been influenced by the training background and athletic characteristics of the participants [10, 11]. Therefore, an investigation using a larger number of participants is necessary to evaluate the relationship between the IRFD measured using ILP and dynamic motor performance. Besides, the IRFD in aforementioned studies was quantified as the steepest slope of the force-time curve during the test, suggesting that the previous studies evaluated only one aspect of the IRFD during the ILP. In addition, the PF during the IRFD test was correlated with CMJ-JH in the previous study [11], indicating a possible influence of the PF on the relationship between IRFD and the CMJ-JH. Therefore, relationships between IRFDs during ILP and dynamic exercise performances remain unclear in terms of multi-phase and relative IRFDs standardized using the PF during the test.

Among dynamic exercise performances, CMJ and rebound continuous jump (RJ) are easy to perform with low risks of injuries compared to the change of direction, sprint, and running jump. Moreover, these vertical jump (VJ) performances are strongly related to various actual sports performances (e.g. 100 m sprint time, agility test time, and players' performance levels [12]) and have broadly been used by coaches and scientists to evaluate the leg-extension force-production capability. Furthermore, the CMJ and the RJ have been categorized as slow stretch-shortening cycle (SSC) and fast SSC exercises, respectively. These two jumps differ in terms of neurophysiological and mechanical aspects [13]. Therefore, examining relationships between multi-phase IRFDs from onset to the terminal phase of force-time curve during ILP with CMJ and RJ performances might improve our understanding of the characteristics of IRFDs during ILP in terms of their relationship with dynamic exercise performances.

The aim of this study was to elucidate characteristics of explosive force-production capabilities represented by multi-phase IRFDs measured using ILP by investigating the relationships of RFD with CMJ and RJ performances. The hypothesis herein was that, when controlling for PF, the relationships between IRFDs and VJ variables vary between the used duration ranges of RFD calculation. Understanding the relationships between IRFDs and VJ performances would help coaches and athletes predict training issues (e. g. motor skills or strength) when planning strength training and effectively improve exercise performance and rehabilitation.

## Materials and methods

### Participants

Two-hundred-and-thirty male collegiate athletes who specialized in soccer, volleyball, basketball, handball, athletics, tennis, badminton, and Japanese martial arts (mean ± SD: age, 20.0 ± 1.1 years; stature, 174.7 ± 7.1 cm; body mass, 70.9 ± 9.3 kg) volunteered to participate in this study. Prior to the experiments, the participants were informed of the purpose, methods, risks, and safety precautions accompanying the experiment. The participants agreed to participate in the experiment after understanding its contents, and written informed consent was obtained. This study was approved by the Research Ethics Committee of the Faculty Health and Sports Science, University of Tsukuba. (tai 30–142).

### Experimental protocol

After a self-selected warm up, the participants (wearing their own shoes) performed an isometric single-leg press (ISLP) and two VJ tests in a single session in random order. For ISLP, IRFD and by unilateral task tends to be greater than the force per leg during bilateral force

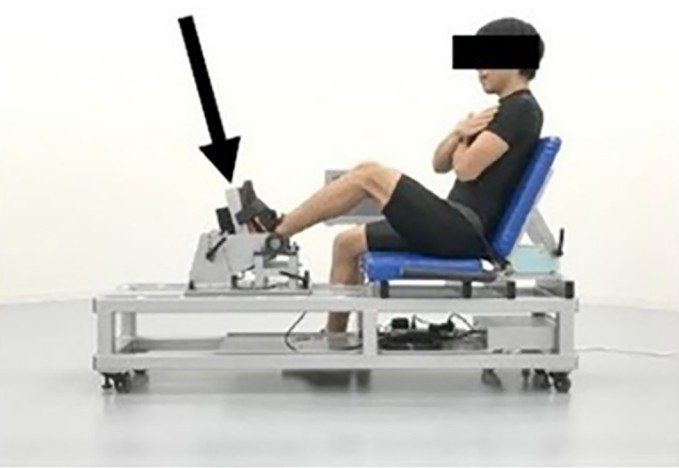

**Fig 1. Custom-built dynamometer: The multi-strength tester and measuring posture of the participant.** Black arrow indicates the load sell.

production [14] furthermore, participants also generate greater force more easily in a unilateral task compared to a bilateral task because participants can focus their attention on a test leg. Taking these assumptions into consideration, leg-extension force-production capability in present study was measured at a single-leg. The ISLP was performed for two trials with the dominant leg on a custom-built dynamometer (multi strength tester, MST [S13158, Takei scientific instruments, Tokyo, Japan; 1000 Hz]). The rest period between the ISLP trials was one minute. The MST was a specific device to measure leg-extension force production and consisted of a foot pedal and seat as shown in Figs 1 and 2. The foot pedal and seat angles from horizontal line of the MST were 75° and 115° respectively. The MST can be used to measure the force applied onto the foot pedal by the strain gauge load cell attached behind the foot

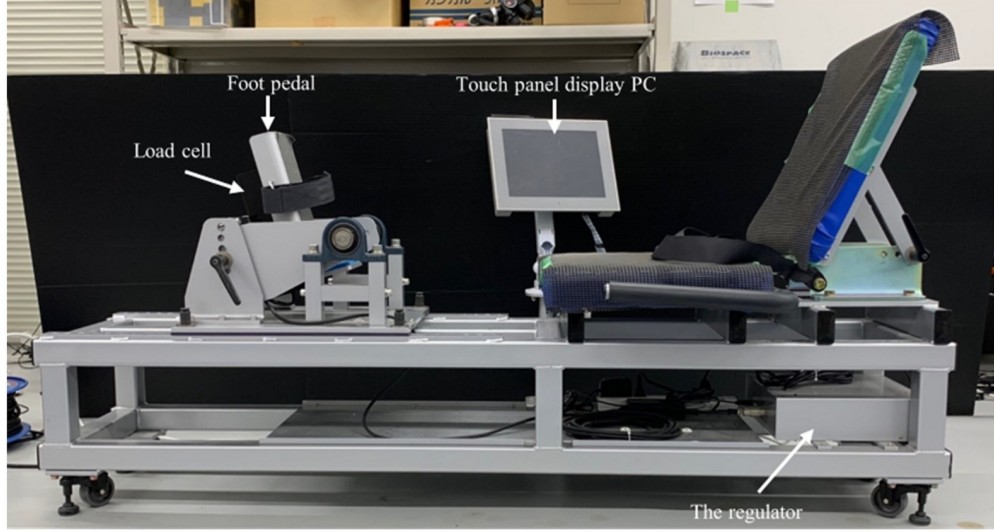

**Fig 2. The view and outside drawing from the side of the custom-built dynamometer (multi-strength tester).**

pedal (C2D1-800K, Minebea, Tokyo, Japan [capacity, 8000 N; non-linearity, hysteresis, and repeatability, 0.018%]). During the ISLP test, the participant sat on the MST seat with a knee joint angle of approximately 115˚ and their low-back against the seat back. The participant's pelvis and foot were respectively stabilized to the seat and foot pedal using adjustable belts. Participants were instructed not to use hip abduction movements or counter movements of leg extension, and the participant was retested if these movements were observed during the trial. A goniometer (SG150, Biometrics, Ladysmith, WI, USA) was used to measure the knee joint angle. With the intent to minimize compliance, participants were instructed to apply approximately 100 N of pre-tension onto the foot pedal while monitoring the force signal on the MST display. Participants were then instructed to produce maximum force against the footplate as quickly as possible. The 100 N of pre-tension was less than 10% of the PF during the ISLP test and had negligible effect on the IRFD [15]. The participants kept the appropriate pre-tension for approximately three seconds before the maximal effort force production, which was initiated by the verbal cue of an experimenter. Participants were instructed to maintain voluntary contraction against the pedals. The duration of the maximal effort contraction was at least three seconds.

For the CMJAS, participants were instructed to jump as high as possible with self-selected countermovement. For the RJAS, which consisted of five continuous jumps, the participants were instructed to jump as high and quickly (minimizing the ground contact time [CT]) as possible. The IRFD measurement was performed using a single-leg, while the VJs were measured using a bilateral leg. A previous study has suggested that a single-leg VJ had a longer movement duration than a double-leg VJ [16] and that movements on the frontal plane significantly influence performance in single-leg VJ [17]. These characteristics of the single-leg VJ suggest that it is a more technically complex than the double-leg VJ and is thus more likely to reflect anything other than the sagittal leg-extension explosive force. Therefore, double-leg VJ is more easily task. Taking the assumption into consideration, VJs in present study was measured at the bilateral task. Pfeile et al. [18] reported no differences in the correlation coefficients between unilateral knee-extension strength and bilateral exercise-performances compared to the correlation coefficient between bilateral knee-extension strength and bilateral exercise-performances. The participants were allowed to swing the arms (Vertical jump with arms swing [VJAS]) for inducing the best jump performances. A lower height than that during their practice was considered a failed attempt. The participants performed each of CMJAS and RJAS on two force platforms (90 cm × 60 cm [type 9287C, Kistler, Amherst, New York, USA]) until at least two successful trials were obtained for each. Participants were instructed to place each foot separately on the two force plates and if the ground contact of the jump was outside each force plate, it was considered a failed attempt. In addition, participants were instructed to keep the legs straight in the air and attempt to land in the same position as that during take-off. If a participant's posture changed significantly during the flight phase, it was considered a failed attempt. All trials were recorded on video to confirm that VJAS had been performed correctly.

The rest period between jump tests was one minute. Prior to ISLP and jump tests, the participants practiced the experimental test exercises at least three times while monitoring the performance variables to familiarize themselves with the tests. The rest period was > 5 min between ISLP and VJAS tests.

## Data processing

Force signals measured using MST were smoothed by a fifth-order zero-lag Butterworth low-pass filter with a cut-off frequency of 25 Hz [19]. The onset of force production was identified

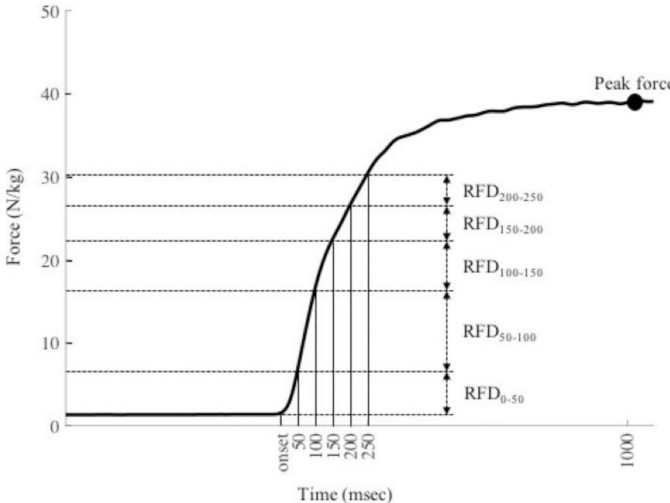

**Fig 3. An example of a force-time curve during isometric single-leg press (ISLP) and IRFDs (IRFD$_{0-50}$, IRFD$_{50-100}$, IRFD$_{100-150}$, IRFD$_{150-200}$, and IRFD$_{200-250}$) and peak force.**

as the instant when a change in the smoothed force signal exceeding 2 N was maintained for 2 ms. The value of the force at the onset was used to standardize the force signal (i.e. approximately 100 N was subtracted from the force signal). The trial with the largest mean IRFD during the entire force-production duration of ISLP was used for further analyses. The entire force-production duration was from onset to the instant when the rate of applied force change fell below zero, and the mean IRFD was calculated by averaging IRFD during the force-production duration. The IRFDs were obtained as rates of increases in the force across 50 ms in the interval from the onset to 250 ms as shown in Fig 3 (IRFD$_{0-50}$, IRFD$_{50-100}$, IRFD$_{100-150}$, IRFD$_{150-200}$, and IRFD$_{200-250}$) [4, 5]. The PF was the highest force recorded in the force-time curve. The IRFDs and PF were normalized to body mass.

For calculations of VJAS test variables, the force platform was used to measure the vertical ground reaction forces at 1000 Hz for each leg. The ground reaction force from the force platform for the dominant leg was used. The takeoff and landing instants of the jump were determined by 10 N vertical force. The jump height (JH) was calculated using a standard method involving total time in the air and free-fall formula [20–23], as follows: jump height = (g · t$_{air}^2$0029·8$^{-1}$, with "g" as the gravitational acceleration with a value of 9.81 m/s$^2$ and "t$_{air}$" as the total time in the air during VJAS. The CMJAS trial with the highest jump height and the RJAS trial with the highest RJAS reactive strength index (RJAS-RSI), which was calculated by dividing RJAS height by the CT, were used for further analysis.

## Statistical analysis

Between-trial reliability of the ISLP measurements was assessed by intra-class correlation coefficients (ICCs), coefficient of variation (CV), and Spearman's rank correlation between-trials. An ICC was considered significant at p < 0.05. For CMJAS and RJAS in the current study, a standard measurement was used with high reliability and low validity, as reported through VJ measurements in previous studies [20–23].

Because IRFD$_{0-50}$, PF, and VJ variables exhibited non-normal distributions as demonstrated using Kolmogorov–Smirnov tests (p ≥ 0.05), non-parametric analyses were performed.

Relationships between IRFDs, PF, and VJAS performances were analyzed by Spearman's rank correlation coefficient. Partial rank correlation coefficient was used to examine the relationships between IRFDs and VJAS performances controlling the influence of PF. The statistical significance was set at $p < 0.05$. Rank correlation coefficients were considered as an effect size, which was translated as small ($0.1 \leq |\rho| < 0.3$), moderate ($0.3 \leq |\rho| < 0.5$), large ($0.5 \leq |\rho| < 0.7$), very large ($0.7 \leq |\rho| < 0.9$), and extremely large ($0.9 \leq |\rho|$) [24]. Statistical analyses were performed using SPSS software (Version 26.0, IBM, Armonk, New York, USA).

## Results

Table 1 summarizes the medians of isometric variables during ISLP and VJAS. The $IRFD_{50-100}$ showed the largest IRFD value. Table 2 summarizes the Spearman's rank correlations between IRFDs, PF, and pre-tension during the single-leg press. The PF was correlated to a small effect with $IRFD_{0-50}$ ($\rho = 0.29$, $p < 0.001$). However, large correlations were noted between PF and $IRFD_{50-100}$ and $IRFD_{100-150}$ ($\rho = 0.55$ and $0.68$, $p < 0.001$; large effect); PF was correlated to a very large effect with $IRFD_{150-200}$ and $IRFD_{200-250}$ ($\rho = 0.72$ and $0.75$, $p < 0.001$). Pre-tension was correlated with $IRFD_{0-50}$ ($\rho = -0.22$, $p = 0.001$; small effect), $IRFD_{50-100}$ ($\rho = -0.18$, $p = 0.006$, small effect), and $IRFD_{150-200}$ to $IRFD_{200-250}$ ($\rho = 0.13$, $p = 0.043$, and $0.15$, $p = 0.027$; small effect).

Table 3 indicates that the means of intra-class ICC, CV, and Spearman's rank correlation between trial measurements of ISLP strength variables had good between-trail reliability, as demonstrated by the significant ICCs. All of ICCs were higher than 0.88, and the lower limits of 95% CI were higher than 0.85. During ISLP, the between-trial CV was largest at $IRFD_{150-200}$ (10.92%) and was the lowest for PF (4.86%). All IRFDs and PF showed extremely large correlation coefficients between two trials ($0.9 \leq \rho$).

Table 4 presents Spearman's rank correlations between isometric variables during ISLP and VJAS. $IRFD_{0-50}$, $IRFD_{50-100}$, and $IRFD_{100-150}$ were correlated to a moderate effect with CMJAS-JH ($\rho = 0.32$, $0.45$, and $0.33$, $p < 0.001$), RJAS-JH ($\rho = 0.39$, $0.46$, and $0.33$, $p < 0.001$), and RJAS-RSI ($\rho = 0.43$, $0.48$, and $0.35$, $p < 0.001$). The $IRFD_{150-200}$ was correlated to a small to moderate effect with CMJAS-JH ($\rho = 0.24$, $p < 0.001$), RJAS-JH ($\rho = 0.25$, $p < 0.001$), and RJAS-RSI ($\rho = 0.32$, $p < 0.001$). The $IRFD_{200-250}$ was correlated to a small effect with

**Table 1. Median, quartile deviation, minimum, maximum, and 95% CI for counter movement jump, rebound continuous jump, and isometric single-leg press variables.**

|  | Measurements | Median | QD | Min | Max | 95% CI | |
|---|---|---|---|---|---|---|---|
|  |  |  |  |  |  | Lower | Upper |
| ISLP | $IRFD_{0-50}$ ($\Delta N \cdot kg^{-1}$) | 3.88 | 1.60 | 1.52 | 8.71 | 3.69 | 4.12 |
|  | $IRFD_{50-100}$ ($\Delta NFF65kg^{-1}$) | 6.68 | 2.34 | 2.20 | 11.57 | 6.45 | 6.97 |
|  | $IRFD_{100-150}$ ($\Delta N \cdot kg^{-1}$) | 4.51 | 1.76 | 0.66 | 8.71 | 4.34 | 4.71 |
|  | $IRFD_{150-200}$ ($\Delta N \cdot kg^{-1}$) | 3.00 | 1.44 | −0.44 | 6.70 | 2.77 | 3.14 |
|  | $IRFD_{200-250}$ ($\Delta N \cdot kg^{-1}$) | 2.66 | 1.44 | −0.47 | 6.29 | 2.44 | 2.81 |
|  | Peak force ($N \cdot kg^{-1}$) | 28.70 | 9.85 | 12.5 | 52.2 | 27.2 | 29.9 |
| CMJAS | Jump height (m) | 0.459 | 0.102 | 0.302 | 0.686 | 0.447 | 0.471 |
| RJAS | Jump height (m) | 0.401 | 0.106 | 0.220 | 0.657 | 0.390 | 0.412 |
|  | Contact time (s) | 0.176 | 0.024 | 0.132 | 0.284 | 0.172 | 0.176 |
|  | RSI ($m \cdot s^{-1}$) | 2.27 | 0.74 | 1.12 | 4.19 | 2.18 | 2.36 |

ISLP (isometric single-leg press), 95% CI (95% confidence interval), CMJAS (countermovement jump with an arm swing), RJAS (rebound continuous jump with an arm swing), RSI (Reactive strength index), QD (quartile deviation), IRFD (isometric rate of force development)

**Table 2. Spearman's rank correlations between the isometric rate of force development, peak force, and pre-tension during single-leg press (n = 230).**

| Variables | PF ($\Delta N \cdot kg^{-1}$) | | Pre-tension ($N \cdot kg^{-1}$) | |
|---|---|---|---|---|
| | ρ (95% CI) | P | ρ (95% CI) | p |
| $IRFD_{0-50}$ ($\Delta N \cdot kg^{-1}$) | **0.29 (0.16–0.40)** | < 0.001 | **−0.22(−0.34–0.09)** | = 0.001 |
| $IRFD_{50-100}$ ($\Delta N \cdot kg^{-1}$) | **0.55 (0.44–0.64)** | < 0.001 | **−0.18(−0.30–0.05)** | = 0.006 |
| $IRFD_{100-150}$ ($\Delta N \cdot kg^{-1}$) | **0.68 (0.59–0.75)** | < 0.001 | −0.05(−0.17-−0.08) | = 0.482 |
| $IRFD_{150-200}$ ($\Delta N \cdot kg^{-1}$) | **0.72 (0.64–0.78)** | < 0.001 | **−0.13(−0.00-−0.26)** | = 0.043 |
| $IRFD_{200-250}$ ($\Delta N \cdot kg^{-1}$) | **0.75 (0.68–0.81)** | < 0.001 | **−0.15(−0.02-−0.27)** | = 0.027 |

IRFD (isometric rate of force development during single-leg press), PF (isometric peak force during single-leg press), 95% CI (95% confidence interval). Bold indicates significant coefficient at p < 0.05.

**Table 3. Intra-class correlation coefficients (ICCs), average between-trial coefficient of variation (CV), and Spearman's rank correlation between two trials for repeated measures of isometric rate of force developments (IRFD) and peak force during isometric leg press.**

| Variables | ICC (95% CI) | CV (%) | ρ (95% CI) |
|---|---|---|---|
| $IRFD_{0-50}$ ($\Delta N \cdot kg^{-1}$) | **0.92 (0.90–0.94)** | 6.09 | **0.92 (0.89–0.94)** |
| $IRFD_{50-100\ 0}$ ($\Delta N \cdot kg^{-1}$) | **0.92 (0.90–0.94)** | 5.93 | **0.93 (0.90–0.95)** |
| $IRFD_{100-150}$ ($\Delta N \cdot kg^{-1}$) | **0.89 (0.86–0.92)** | 8.15 | **0.91 (0.88–0.93)** |
| $IRFD_{150-200}$ ($\Delta N \cdot kg^{-1}$) | **0.88 (0.85–0.91)** | 10.92 | **0.91 (0.88–0.93)** |
| $IRFD_{200-250}$ ($\Delta N \cdot kg^{-1}$) | **0.89 (0.86–0.92)** | 6.58 | **0.92 (0.89–0.94)** |
| Peak force ($N \cdot kg^{-1}$) | **0.95 (0.93–0.96)** | 4.86 | **0.95 (0.93–0.96)** |

95% CI (95% confidence interval). Bold indicates significant ICC and significant coefficient at p < 0.05.

**Table 4. Spearman's rank correlations between isometric variables during single-leg press and vertical jump variables (n = 230).**

| Variables | CMJAS | | RJAS | | | | | |
|---|---|---|---|---|---|---|---|---|
| | JH (m) | | JH (m) | | CT (s) | | RSI ($m \cdot s^{-1}$) | |
| | ρ (95% CI) | P | ρ (95% CI) | P | ρ (95% CI) | P | ρ (95% CI) | p |
| $IRFD_{0-50}$ ($\Delta N \cdot kg^{-1}$) | **0.32 (0.19–0.43)** | < 0.001 | **0.39 (0.27–0.49)** | < 0.001 | **−0.29 (−0.41-−0.17)** | < 0.001 | **0.43(0.32–0.54)** | < 0.001 |
| $IRFD_{50-100}$ ($\Delta N \cdot kg^{-1}$) | **0.45 (0.33–0.55)** | < 0.001 | **0.46 (0.34–0.56)** | < 0.001 | **−0.24 (−0.36-−0.11)** | < 0.001 | **0.48(0.37–0.58)** | < 0.001 |
| $IRFD_{100-150}$ ($\Delta N \cdot kg^{-1}$) | **0.33 (0.21–0.45)** | < 0.001 | **0.33 (0.20–0.44)** | < 0.001 | **−0.15 (−0.27-−0.02)** | = 0.025 | **0.35(0.23–0.46)** | < 0.001 |
| $IRFD_{150-200}$ ($\Delta N \cdot kg^{-1}$) | **0.24 (0.11–0.36)** | < 0.001 | **0.25 (0.13–0.37)** | < 0.001 | **−0.25 (−0.37-−0.13)** | < 0.001 | **0.32(0.20–0.44)** | < 0.001 |
| $IRFD_{200-250}$ ($\Delta N \cdot kg^{-1}$) | **0.20 (0.07–0.32)** | = 0.003 | **0.22 (0.09–0.34)** | = 0.001 | **−0.22 (−0.34-−0.09)** | = 0.001 | **0.29(0.16–0.40)** | < 0.001 |
| PF ($N \cdot kg^{-1}$) | **0.30 (0.17–0.41)** | < 0.001 | **0.36 (0.24–0.47)** | < 0.001 | **−0.29 (−0.40-−0.16)** | < 0.001 | **0.43(0.31–0.53)** | < 0.001 |

IRFD (isometric rate of force development during single-leg press), PF (isometric peak force during single-leg press), CMJAS (countermovement jump with an arm swing), RJAS (rebound continuous jump with an arm swing), JH (jump height), CT (contact time), RSI (reactive strength index). 95% CI (95% confidence interval). Bold indicates significant coefficient at p < 0.05.

CMJAS-JH (ρ = 0.20, p = 0.003), RJAS-JH (ρ = 0.22, p = 0.001), and RJAS-RSI (ρ = 0.29, p < 0.001). In addition, RJAS-CT was correlated to a small effect with all periods of IRFD (−0.29 < ρ < −0.15, p ≤ 0.025). For the PF, significant correlations were found between CMJAS-JH, RJAS-JH, RJAS-CT, and RJAS-RSI (ρ = 0.30, 0.36, −0.29, and 0.43, respectively, p <0.001; small to moderate effects).

**Table 5. Partial rank correlations between isometric rate of force development during single-leg press and vertical jump variables controlling for isometric peak force during single-leg press (n = 230).**

| Variables | Control variable | CMJAS | | RJAS | | | | | |
|---|---|---|---|---|---|---|---|---|---|
| | | JH (m) | | JH (m) | | CT (s) | | RSI (m·s⁻¹) | |
| | | $\rho_{xy/z}$ | P | $\rho_{xy/z}$ | p | $\rho_{xy/z}$ | P | $\rho_{xy/z}$ | p |
| IRFD$_{0-50}$ ($\Delta$N·kg⁻¹) | PF | **0.26** | **< 0.001** | **0.32** | **< 0.001** | **−0.23** | **< 0.001** | **0.36** | **< 0.001** |
| IRFD$_{50-100}$ ($\Delta$N·kg⁻¹) | | **0.36** | **< 0.001** | **0.33** | **< 0.001** | −0.10 | = 0.129 | **0.33** | **< 0.001** |
| IRFD$_{100-150}$ ($\Delta$N·kg⁻¹) | | **0.19** | **= 0.004** | 0.12 | = 0.077 | −0.07 | = 0.312 | 0.09 | = 0.173 |
| IRFD$_{150-200}$ ($\Delta$N·kg⁻¹) | | 0.04 | = 0.526 | −0.01 | = 0.869 | −0.06 | = 0.375 | 0.03 | = 0.637 |
| IRFD$_{200-250}$ ($\Delta$N·kg⁻¹) | | −0.04 | = 0.173 | −0.09 | = 0.173 | −0.01 | = 0.871 | −0.05 | = 0.428 |

IRFD (isometric rate of force development during single-leg press), PF (isometric peak force during single-leg press), CMJAS (countermovement jump with an arm swing), RJAS (rebound jump with an arm swing), JH (jump height), CT (contact time), RSI (reactive strength index). Bold indicates significant coefficient at $p < 0.05$

Table 5 presents partial rank correlations between IRFDs and VJAS variables. When controlling for the influence of PF, IRFD$_{0-50}$ was correlated with all VJAS variables (CMJAS-JH: $\rho_{xy/z} = 0.26$, p < 0.001; RJAS-JH: $\rho_{xy/z} = 0.32$, p < 0.001; RJAS-CT: $\rho_{xy/z} = −0.23$, p < 0.001; RJAS-RSI: $\rho_{xy/z} = 0.36$, p < 0.001), while IRFD$_{50-100}$ was correlated with CMJAS-JH, RJAS-JH, and RJAS-RSI ($\rho_{xy/z} = 0.36$, $\rho_{xy/z} = 0.33$, and $\rho_{xy/z} = 0.33$, respectively, p < 0.001) without RJAS-CT ($\rho_{xy/z} = −0.10$, p = 0.129). Moreover, when controlling for the influence of PF, IRFD$_{100-150}$ was only correlated with CMJAS-JH ($\rho_{xy/z} = 0.19$, p = 0.004).

## Discussion

This study aimed to elucidate the characteristics of the explosive force-production capabilities represented by multi-phase IRFDs measured using ISLP by investigating the relationships with VJAS performances. On controlling the influence of PF, IRFDs during the early phase of the force-time curve (IRFD$_{0-50}$, IRFD$_{50-100}$, and IRFD$_{100-150}$) were associated with VJAS performances in 230 athletes included in this study. In addition, RJAS variables were only associated with IRFDs in the earlier phase (IRFD$_{0-50}$ and IRFD$_{50-100}$), unlike the associations between CMJAS-JH and IRFDs (IRFD$_{0-50}$, IRFD$_{50-100}$, and IRFD$_{100-150}$). Although this study found significant relationships, the magnitude of the correlations was not high (< 0.48 for simple rank correlation), which should be considered when translating the current findings into practice.

Small rank correlation coefficients were found between the magnitude of pre-tension and IRFDs during the early phase of contraction (Table 2). Pre-tension is known to change the discharge pattern of the motor unit during explosive contraction and to decrease IRFD [14]. Therefore, IRFDs can be underestimated in individuals with excessive pre-tension. However, in this study, IRFDs and PF showed higher ICC and lower CV (Table 3) than those reported in previous studies [25–27] that used isometric contraction. Moreover, extremely large correlation coefficients between two ISLP trails were observed. de Ruiter [28] suggested that unreliable agonist muscle activities prior to explosive-isometric contraction decrease the reliability of IRFD during the early phase. Hence, in the current study, high reliability of the IRFDs may be attributed to the fact that the pre-tension controlled agonist muscle activities prior to force onset. From the perspective of the participants, the utilization of standardized pre-tension may have facilitated the establishment of a stable baseline prior to force production. This could have potentially mitigated uncontrolled countermovement, excessive pre-tension, and slack in body and equipment, thereby enhancing the reliability of IRFD measurements [3]. Therefore,

IRFD measurement in this study is a reliable method of assessing the explosive force-production capability in multi-joint exercise.

When controlling for the influence of PF, all partial rank correlation coefficients between IRFDs and VJAS variables were lower than the corresponding simple rank correlation coefficients (Tables 4 & 5). These results demonstrate that the PF affecting relationships between IRFDs were calculated from onset to 250 ms of the force-time curve and VJAS variables, which is consistent with the findings of a previous study by Suchomel et al. [12]. When controlling for the PF, IRFDs from onset to 150 ms and onset to 100 ms were correlated with CMJAS-JH and RJAS variables, respectively (Table 5). These partial rank correlation results suggested that IRFDs in the early phase of the force-time curve during ISLP may represent the force-production capability required for VJASs that would not be assessed by PF. The CMJAS and RJAS are categorized as slow and fast SSC exercises, respectively, based on differences in neurophysiological and mechanical aspects [13]. Therefore, it may be possible that the IRFDs calculated from onset to 150 ms in ISLP represent the slow SSC force-production capability, while IRFDs calculated from onset to 100 ms in ISLP represent the fast SSC force-production capability. Alternatively, the highest partial rank correlation coefficients with CMJAS-JH ($\rho_{xy/z} = 0.36$) and RJAS-JH ($\rho_{xy/z} = 0.33$) were found for $IRFD_{50-100}$, and the highest partial rank correlation coefficient with RJAS-RSI was found for $IRFD_{0-50}$ ($\rho_{xy/z} = 0.36$). These results indicate that specific IRFDs demonstrate the strongest relationship with respective VJAS variables. Therefore, multi-phase IRFD measurements calculated from onset to 150 ms in a seated position, like the ISLP, may allow us to evaluate leg-extension force-production capabilities from multiple perspectives. Given the advantage of the multi-phase IRFDs measured in a seated position (easier and safer than IMTP and ISq), an evaluation of leg (mainly hip and knee) extension strength characteristics by multi-phase IRFDs measured using ISLP would be useful for monitoring athlete preparation and training adaptation levels from multiple aspects.

The highest correlation coefficient between $IRFD_{50-100}$ measured at a knee joint angle of 115˚ and CMJAS-JH in the current study ($\rho = 0.45$ [95% CI: 0.33–0.55]) was lower than that in a comparable previous study ($r = 0.69$), which used the maximal RFD during the ILP measured at a knee joint angle of 120˚ [10]. This contradiction may have resulted due to the difference in characteristics and number of participants (14 active male postgraduate students in the previous study versus 230 well trained athletes in this study) and the slight difference in knee joint angle (120˚ in the previous study [10] versus 115˚ in this study). The corresponding correlation coefficient using the RFD at a knee joint angle of 90˚ was 0.37 in the same previous study [10], indicating that a smaller knee joint angle during ILP could result in a weaker relationship with CMJ-JH. The ratio of fast-twitch fibers [4, 5, 29] and the stiffness [29, 30] of the vastus lateralis are determinants of both CMJ-JH and maximal IRFD during an isometric knee extension test. Moreover, de Ruiter et al. [28] suggested that a feedforward regulation of the vastus lateralis muscle contraction influences both CMJ-JH and IRFD during a knee extension exercise. Taken together, the above common neuromuscular and muscle composition factors, which affect both knee extension IRFDs and CMJ, could result in the highest simple rank correlation coefficient between $RFD_{50-100}$ and CMJAS-JH. The $IRFD_{200-250}$ showed a significant but weak simple rank correlation ($\rho = 0.20$: [95% CI: 0.07–0.32], small effect) with CMJAS-JH, and the partial rank correlation coefficients between IRFDs after 150 ms and CMJAS-JH were not significant. Although it is difficult to provide a clear explanation, this weak or lack of relationship could possibly result from the relatively small inter-individual variability of $IRFD_{200-250}$ compared to IRFDs in early phases. This is likely because the force-time curve during the later phase was shallower and stable, and the force-time curve leveled off after 200 ms for some participants in this study.

The significant partial rank correlations between IRFDs during the ISLP and RJAS variables were found in the range of the force-time curve of shorter duration than the corresponding range for the relationship with CMJAS-JH, and the magnitudes of partial rank correlations were comparably not strong (Table 5). These results could be a consequence of the RJ requiring greater force production within a shorter duration compared to the CMJ [13, 31]. The highest partial rank correlation coefficient ($\rho_{xy/z} = 0.33$) of IRFDs with RJAS-JH was found for $IRFD_{50-100}$, while the magnitude of the coefficient was slightly smaller than the corresponding value using CMJAS-JH ($\rho_{xy/z} = 0.36$), indicating that $IRFD_{50-100}$ can be superior for evaluating CMJAS-JH than RJAS-JH. Horita et al. [32] reported that the maximal IRFD during an isometric knee extension test was correlated with the knee extension moment at the end of the braking phase during a drop jump, while the corresponding knee extension moment was correlated with takeoff velocity during the drop jump, which may determine RJ-JH. As the $IRFD_{50-100}$ was the highest IRFD in this study (i.e. maximal IRFD), the aforementioned findings in the previous study may explain the highest partial rank correlation coefficient of RJAS-JH with $IRFD_{50-100}$ in this study. This suggests that the greater maximal IRFD during the ISLP is likely responsible for the greater RJAS-JH. Only the IRFDs during the first 100 ms were related to RJAS-RSI. Horita et al. [32] suggested that increasing knee-joint stiffness through an acute increase in knee extension moment in the braking phase of the drop jump contributes to achieving a high takeoff velocity and short CT. Because high takeoff velocity and short CT result in high RSI values, IRFD measured by ISLP could be an indicator of knee-joint stiffness that contributes to fast SSC ability such as RJAS. In contrast to its relationship with RJAS-JH, the highest partial rank correlation coefficient with RJAS-RSI was found for $IRFD_{0-50}$. The magnitude of the RJAS-RSI was influenced by the fact that the RJAS-CT was only correlated with $IRFD_{0-50}$ when controlling the influence of PF (Tables 4 & 5).

For the fast SSC exercises, such as the RJAS, a suitable activity level and timing of agonist muscle activation during pre-landing contributed to the production of a large amount of mechanical power by leg extension during the following support phase [32, 33]. Pre-landing activation occurs within a very short time just before touchdown, and the neural drive from the central nervous system and characteristics of excitation-contraction coupling affect the pre-landing conditions [33, 34]. The characteristics of excitation-contraction coupling were suggested to be a determinant factor of the knee extension IRFD from onset to 50 ms [5]. Hence, $IRFD_{0-50}$ might possibly relate to skillful landing in RJAS. Consequently, the significant correlation of $IRFD_{0-100}$ and RJ variables suggested that IRFD from onset to 100 ms may represent capabilities of an explosive force production during the breaking phase of the fast SSC exercise, such as the RJAS.

The practical implications of the results in this study are recommended that IRFDs at the steepest phase of the force-time curve (50–100 ms) should be measured to assess the leg-extension capabilities for slow SSC, while IRFDs during the early phase of the force-time curve (0–50 ms) should be measured to assess the leg-extension capabilities for fast SSC. However, a decrease in partial rank correlation coefficients between IRFDs and VJAS performances suggests that both PF and IRFDs should be evaluated in the training. Moreover, perhaps IRFDs should be evaluated their absolute values without normalization by PF.

The current study participated athletes from various sports, with force-production capabilities were measured unilaterally, while VJAS were measured bilaterally. In addition, VJAS were performed without setting joint-angles condition. Variations in sports-specific characteristics among participants influence extent of bilateral deficit, joint angle-force property and jump technique, potentially causing certain individuals to either overestimating or underestimating their VJ and/or ISLP performances. In further research, it is necessary to employ

corresponding measurement position/setups for VJAS and IRFD-measurement, while recruiting participants with homogeneous sports-specific characteristics.

## Conclusion

IRFDs measured by ISLP from onset to 150 ms had a relatively smaller influence on PF compared to IRFDs after 150 ms, indicating a direct relationship between IRFDs calculated from onset to 150 ms and VJAS performances. Moreover, while IRFDs from onset to 150 ms might be indicative of the muscular aspect of CMJAS-JH, IRFDs calculated from onset to 100 ms might be a limiting or determining factor of RJAS variables (RJAS-JH, RJAS-CT, and RJAS-RSI). Thus, the early-phase IRFD measured by ISLP could enable the assessment of multiple aspects of leg-extension strength characteristics that are different from maximal strength. The findings of this study will provide valuable insights for coaches and athletes when using IRFDs to evaluate the adaptation level of leg-extension strength for strength training for explosive performance capabilities.

## Supporting information

**S1 File.**
(PDF)

## Acknowledgments

The authors thank the participants for their valuable contribution to this study.

## Author Contributions

**Conceptualization:** Kodayu Zushi, Ryu Nagahara, Amane Zushi.

**Data curation:** Kodayu Zushi, Takuya Yoshida, Amane Zushi.

**Formal analysis:** Kodayu Zushi, Takuya Yoshida, Amane Zushi.

**Funding acquisition:** Kodayu Zushi.

**Investigation:** Kodayu Zushi, Takuya Yoshida, Amane Zushi.

**Methodology:** Kodayu Zushi, Yasushi Kariyama, Ryu Nagahara, Keigo Ohyama-Byun.

**Project administration:** Keigo Ohyama-Byun, Mitsugi Ogata.

**Resources:** Kodayu Zushi, Ryu Nagahara, Keigo Ohyama-Byun.

**Software:** Kodayu Zushi.

**Supervision:** Keigo Ohyama-Byun, Mitsugi Ogata.

**Validation:** Kodayu Zushi, Yasushi Kariyama, Ryu Nagahara, Takuya Yoshida.

**Visualization:** Kodayu Zushi, Yasushi Kariyama, Ryu Nagahara, Takuya Yoshida.

**Writing – original draft:** Kodayu Zushi, Yasushi Kariyama, Ryu Nagahara, Takuya Yoshida, Amane Zushi, Keigo Ohyama-Byun, Mitsugi Ogata.

**Writing – review & editing:** Kodayu Zushi, Yasushi Kariyama, Ryu Nagahara, Takuya Yoshida, Amane Zushi, Keigo Ohyama-Byun, Mitsugi Ogata.

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
