## [Decision Letter · Decision Letter 0]

17 Jun 2022

PONE-D-21-32499Association of multi-phase rates of force development during an isometric leg press with vertical jump performancesPLOS ONE

Dear Dr. Zushi,

Thank you for submitting your manuscript to PLOS ONE. After careful consideration, we feel that it has merit but does not fully meet PLOS ONE’s publication criteria as it currently stands. Therefore, we invite you to submit a revised version of the manuscript that addresses the points raised during the review process.

Please see the comments from one reviewer below. Please note that we have only been able to secure a single reviewer to assess your manuscript. We are issuing a decision on your manuscript at this point to prevent further delays in the evaluation of your manuscript. Please be aware that the editor who handles your revised manuscript might find it necessary to invite additional reviewers to assess this work once the revised manuscript is submitted. However, we will aim to proceed on the basis of this single review if possible.  In addition, we ask that you provide the full name of the IRB in the Methods section.

We look forward to receiving your revised manuscript.

Kind regards,

Hanna Landenmark

Staff Editor

PLOS ONE

**Journal requirements:**

“No”

d) If you did not receive any funding for this study, please state: “The authors received no specific funding for this work.

Reviewers' comments:

Reviewer's Responses to Questions

**Comments to the Author**

1. Is the manuscript technically sound, and do the data support the conclusions?

Reviewer #1: Partly

2. Has the statistical analysis been performed appropriately and rigorously? 

Reviewer #1: Yes

3. Have the authors made all data underlying the findings in their manuscript fully available?

Reviewer #1: Yes

4. Is the manuscript presented in an intelligible fashion and written in standard English?

Reviewer #1: No

5. Review Comments to the Author

Reviewer #1: General Comments

The authors have completed a substantial body of work, in a very relevant athlete cohort. While the grammatical and writing style should be improved to ensure that ideas and concepts are presented as succinctly as possible, the major flaw in my opinion is the reliance placed on statistical significance of the correlation r values rather than the strength of these relationships. A stronger manuscript would be positioned around the strength or lack there of for the correlation analysis in such a large homogenous athlete cohort as compared to the very small previous cohorts that reported stronger relationships.

Specific Comment

Ln47-48; Poor grammar

Ln88-94; Please reconsider the sentence length and structure as currently it is grammatically poor

Ln116-118; Single sentence paragraphs are not ideal, I suggest including a statement that succinctly details what the proposed benefit to the area would be if the data is supportive of the hypothesis as method for expanding this paragraph

Ln121; Delete the word 'were'

Ln122-126; Delete the words, 'of the current study'

Ln141-142; Delete the words, 'at the long seat position'

Ln142; Change the 'deg' to the symbol for degrees, change 'the' to 'their'

Ln142-143; Change 'being attached to' to 'against'

Ln146-151; These three sentences have poor grammar and need to be reordered to improve the flow of information and improve the grammar and presentation

Ln154; I believe that the authors are referring to asking the participants to maintain the voluntary contraction against the pedal. Please re-word

Ln158; Figure 1 Caption, it is unclear what the authors are referring to with the word 'allow'

Ln 158-159; Reconsider the use of the terms 'force production' as 'contraction' seems to be a better term to be used

Ln181; Poor grammar, please insert 'by' so the sentence reads, ".....RFD was calculated by averaging...."

Ln190-191; I am uncertain as to why the authors have chosen to make a comparison between a single leg isometric test and a bipedal test dynamic test when the dynamic test was conducted using two force plates. Did the authors consider looking at the force time characteristics from the dominant leg recorded during the dynamic test?

Ln192; The Bosco method is not a correct methodological reference, was this a flight-time calculation or an impulse momentum calculation?

Ln200; I would generally prefer to see a zero before all decimal places but I understand that this may not be the journal convention, please double check the submission guidelines

Ln206-2014; Please indicate for the reader if any of these comparisons were statistically significant

Ln220; Table 2, Please include the confidence interval in this table

Ln233; Table 3, Again it seems strange to have correlated variables from a single leg assessment to variables (data) derived from a bipedal assessment.

Ln233; Table 3, Considering the number of participants, the relationship or lack thereof between IRFD@200-250ms and the CMJ JH is a very important finding

Ln248; Table 4, While some of these partial correlations are statistically significant they are small to moderate at best. Can the confidence interval be included, so the reader can be given an indication as to the spread of the relationship.

Ln272; Change the word 'to' to 'with' so the sentence should read, "... which is consistent with a previous study..."

Ln269-291; Unfortunately again this section is all about making comparisons between a single leg isometric metric compared to a bipedal dynamic assessment. The authors should consider conducting a single v single leg metric comparison.

Ln293; While the authors have hilghghited that the correlation coefficient is line but compared value is substantially higher, please provide a critical comparison and contrast point to this result

Ln304; Unfortunately the authors have relied very heavily on the statistical significance of the correlation outcomes, however a major factor to these significant correlations is the large sample size, the actual r value is moderate at best and should be more strongly reflected in the level o discussion and interpretation throughout the manuscript

Ln309; Please insert the actual r value in brackets after IRFD to provide the reader with immediate context rather than needing to refer to the table

Ln313-316; This sentence has poor grammar please reword,

6. PLOS authors have the option to publish the peer review history of their article (what does this mean?). If published, this will include your full peer review and any attached files.

Reviewer #1: **Yes: **Dale Wilson Chapman

---

## [Author Response · Author response to Decision Letter 0]

27 Aug 2022

Augast 25, 2022

Editorial Board

PLoS ONE

Dear Editors:

I wish to resubmit an original article for publication in PLoS ONE, titled “Association of multi-phase rates of force development during an isometric leg press with vertical jump performances.” The manuscript ID is PONE-D-21-32499 

We thank you and the reviewers for your thoughtful suggestions and insights. The manuscript has benefited from these insightful suggestions. I look forward to working with you and the reviewers to move this manuscript closer to publication in PLoS ONE.

The manuscript has been rechecked, and the necessary changes have been made in accordance with the reviewers’ suggestions. The responses to all comments have been prepared and attached herewith. Selected content from the Supporting Information has been added to the manuscript file as Fig 2. Moreover, We have removed the relevant "Funding" according to your instruction.

All study participants provided informed consent, and the study design was approved by the appropriate ethics review board on April 25, 2019 (approval number, tai 30-142) and we include our ethics statement in the ‘Methods’ section of our manuscript file (L126-131). We have read and understood your journal’s policies, and we believe that neither the manuscript nor the study violates any of these. The authors have declared that no competing interests exist. Amane Zushi　 and Takuya Yoshida who had been responsible for the data collection and analysis was supported by the Japan Society for the Promotion of Science (JSPS) Grant-in-Aid for Scientific Research (Respectively 20K23310 and 21K17564).

Thank you for your consideration. I look forward to hearing from you. The funder (JSPS) had no role in study design, data collection and analysis, decision to publish, or preparation of the manuscript.

Response to the comments

Manuscript ID# PONE-D-21-32499

The title of manuscript: 

Association of multi-phase rates of force development during an isometric leg press with vertical jump performances

Response to Reviewer # 1

General Comments

The authors have completed a substantial body of work, in a very relevant athlete cohort. While the grammatical and writing style should be improved to ensure that ideas and concepts are presented as succinctly as possible, the major flaw in my opinion is the reliance placed on statistical significance of the correlation r values rather than the strength of these relationships. A stronger manuscript would be positioned around the strength or lack there of for the correlation analysis in such a large homogenous athlete cohort as compared to the very small previous cohorts that reported stronger relationships.

[Response]

Thank you very much for taking your time to review our study. We have tried to improve grammar and writing in the manuscript as much as possible. Moreover, we have now focused on the strength of the relationship and discussed the difference compared to previous small cohort studies (L322-270, 391-395). 

Specific Comment

Ln47-48; Poor grammar

[Response]

We have checked and revised the grammatical errors in lines 48–51.

Ln88-94; Please reconsider the sentence length and structure as currently it is grammatically poor

[Response]

We have checked and revised the sentence length and structure in lines 88-91.

Ln116-118; Single sentence paragraphs are not ideal, I suggest including a statement that succinctly details what the proposed benefit to the area would be if the data is supportive of the hypothesis as method for expanding this paragraph

[Response]

Thank you for your suggestion. We have added a note regarding the areas to which the results of this study can be applied if the hypothesis is dictated (L112-119).

Ln121; Delete the word 'were'

[Response]

Thank you for bringing this to our attention. We have removed the relevant text according to your instruction (L122).

Ln122-126; Delete the words, 'of the current study'

[Response]

Thank you for bringing this to our attention. We have removed the relevant text according to your instruction (L126).

Ln141-142; Delete the words, 'at the long seat position'

[Response]

Thank you for bringing this to our attention. We have removed the relevant text according to your instruction (L143-144).

Ln142; Change the 'deg' to the symbol for degrees, change 'the' to 'their'

[Response]

Thank you for bringing this to our attention. We have revised the relevant text according to your instruction (L140, 144-149).

Ln142-143; Change 'being attached to' to 'against'

[Response]

Thank you for bringing this to our attention. We have revised the relevant text according to your instruction (L144).

Ln146-151; These three sentences have poor grammar and need to be reordered to improve the flow of information and improve the grammar and presentation

[Response]

We have corrected the grammar and flow in the areas to which you refer. Please check the revised section (L149-153) .

Ln154; I believe that the authors are referring to asking the participants to maintain the voluntary contraction against the pedal. Please re-word

[Response]

Thank you for bringing this to our attention. We have corrected the sentence accordingly (L155-156).

Ln158; Figure 1 Caption, it is unclear what the authors are referring to with the word 'allow'

[Response]

Thank you for bringing this to our attention. The text to which you refer was a spelling error. Therefore, we have corrected it to “arrow” (L158-159, Fig 1).

Ln 158-159; Reconsider the use of the terms 'force production' as 'contraction' seems to be a better term to be used

[Response]

As you indicated, we changed the term to "contraction" (L156).

Ln181; Poor grammar, please insert 'by' so the sentence reads, ".....RFD was calculated by averaging...."

[Response]

Thank you for the suggestion. We have added "by" (L186).

Ln190-191; I am uncertain as to why the authors have chosen to make a comparison between a single leg isometric test and a bipedal test dynamic test when the dynamic test was conducted using two force plates. Did the authors consider looking at the force time characteristics from the dominant leg recorded during the dynamic test?

[Response]

It is noted that two force plates were used in the method in this study, which is the setup used by us to calculate performance variables and kinetics and kinematics variables for jumping exercises in our experimental facility. Although, we used to measure and process from dominate leg (L146, L195-196), trials in which leg laterality such as ground contact timing during the VJ were significantly were excluded according to the ground reaction forces of the two force plates; hence, CMJ jumping height and RJ jumping height and ground contact time were calculated correctly. Moreover, kinetics and kinematics variables and during VJs had not reported in this study.

The aim of this study was to elicudate the association between isometric RFD during explosive leg extension and dymamic force production capability during leg extension exercise, such as VJ performances, focusing on duration of force production. Therefore, RJ, which has performed within an extremely short duration (approiximately 0.1-0.2 sec), and CMJ (approiximately 0.5-1.0 sec), which has performed within a longer duration are used in the jump exercise. While IRFD is measured from single leg test, we recognize that your comment suggeste that VJ should be measured through a single leg test. However, previous studies were suggested that the contact time of a single leg test is greater than that of a double leg VJ. Further, double leg VJs are affected not only by leg extension, but also by other movements such as about hip joint abduction, which was hardly detectable in double leg VJs (Sado et al., 2020). These characteristics of the single leg VJs suggest that it is a more skilled jump than the double leg VJs, involving many factors other than plain explosive force exertion. For these reasons, we thought that a trial with bilateral jumps would be appropriate to compare with the ISLPs in this study, and the VJs in this study performed with duble leg jumps (L164-167).

Ln192; The Bosco method is not a correct methodological reference, was this a flight-time calculation or an impulse momentum calculation?

[Response]

The VJ variables in this study were calculated from ground contact time and flight time. Moreover, the citation was inappropriate and has been removed (L197-199).

Ln200; I would generally prefer to see a zero before all decimal places but I understand that this may not be the journal convention, please double check the submission guidelines

[Response]

Thank you for bringing this to our attention. We have made the addition without omitting the zero as you indicated. We have also corrected the other statistical results (L207-209).

Ln206-214; Please indicate for the reader if any of these comparisons were statistically significant

[Response]

Thank you for bringing this to our attention. As you indicated, we have added p-values and correlation or partial correlation coefficients for items that were significant in the results. You may review the edits in the Results section (L212-263).

Ln220; Table 2, Please include the confidence interval in this table

Thank you for bringing this to our attention. I have included the 95% confidence interval in Table 2 (L242-252).

Ln233; Table 3, Again it seems strange to have correlated variables from a single leg assessment to variables (data) derived from a bipedal assessment.

[Response]

Thank you for bringing this to our attention. Although this response contains some overlap with the previous answer, this study aimed to elicudate the relationship between IRFD during leg extension and dynamic performances with very short exercise durations, such as VJs. The single leg VJ requires more skill than a double leg VJ, as it involves many factors other than plain explosive force exertion. For this reason, double leg VJs were performed in this study. In addition, we ensured that no hip abduction or similar movements were used in the leg presses performed in this study, and trials in which these movements were observed were not used in the analysis. We have added this to the Methods (L146-148).

Ln233; Table 3, Considering the number of participants, the relationship or lack thereof between IRFD@200-250ms and the CMJ JH is a very important finding

Thank you for bringing this to our attention. As you indicated, we have added a discussion of the relationship between IRFD200-250 and CMJ-JH (L323-329).

Ln248; Table 4, While some of these partial correlations are statistically significant they are small to moderate at best. Can the confidence interval be included, so the reader can be given an indication as to the spread of the relationship.

 [Response]

Thank you for your suggestion.Unfortunately, we could not find the calculation of confidence intervals from partial correlation analysis in the statistical analysis software used for the partial correlation analysis.

Ln272; Change the word 'to' to 'with' so the sentence should read, "... which is consistent with a previous study..."

Thank you for bringing this to our attention. We have corrected it (L288).

Ln269-291; Unfortunately again this section is all about making comparisons between a single leg isometric metric compared to a bipedal dynamic assessment. The authors should consider conducting a single v single leg metric comparison.

[Response]

Thank you for bringing this to our attention. As you point out,It is generally reasonable to examine the rerationship between a single leg IRFD to singl leg VJs performances. In accordance with the answer to your previous comment, the mechanics and complexities of a single leg VJ may prevent a participant from producing explosive leg extension forces; thus, the use of a double leg jump was more appropriate for achieving the purpose of this study. Although kinetic and kinematics variables were not reported in this study, there was no leg laterality to be considered. Therefore, the present study did not consider it problematic to conduct a double leg VJ and to measure IRFD with a single leg trial. 

Ln293; While the authors have hilghghited that the correlation coefficient is line but compared value is substantially higher, please provide a critical comparison and contrast point to this result

[Response]

Thank you for bringing this to our attention. We have added a discussion of the lower correlation coefficients compared with those of previous studies (L308-329).

Ln304; Unfortunately the authors have relied very heavily on the statistical significance of the correlation outcomes, however a major factor to these significant correlations is the large sample size, the actual r value is moderate at best and should be more strongly reflected in the level o discussion and interpretation throughout the manuscript

[Response]

Thank you for bringing this to our attention. The text has been amended to take into account that the correlation coefficient is moderate.

Ln309; Please insert the actual r value in brackets after IRFD to provide the reader with immediate context rather than needing to refer to the table

[Response]

Thank you for bringing this to our attention. We have included the r values (L309, 324, 336, 338).

Ln313-316; This sentence has poor grammar please reword,

[Response]　

We have corrected the grammar and flow in the areas you pointed out (L345-346).

Thank you for reviewing.

Sincerely,

Kodayu Zushi, PhD

Faculty of Economics, Shiga University, Hikone, Shiga, Japan

1-1-1 Banba, Hikone, Shiga, 522-8522 Japan

Tel.: +81-90-8763-9396

Fax: +81-794-27-1124

E-mail: kodayu-zushi@biwako.shiga-u.ac.jp

---

## [Decision Letter · Decision Letter 1]

13 Feb 2023

PONE-D-21-32499R1Association of multi-phase rates of force development during an isometric leg press with vertical jump performancesPLOS ONE

Dear Dr. Zushi,

Thank you for submitting your manuscript to PLOS ONE. After careful consideration, we feel that it has merit but does not fully meet PLOS ONE’s publication criteria as it currently stands. Therefore, we invite you to submit a revised version of the manuscript that addresses the points raised during the review process.

The reviewers have made several comments mainly around clarification of/the methods as well as the interpretation of the findings. I encourage you to consider their thorough comments carefully and amend the manuscript accordingly or provide a robust rebuttal. 

We look forward to receiving your revised manuscript.

Kind regards,

Theodoros M. Bampouras

Academic Editor

PLOS ONE

Reviewers' comments:

Reviewer's Responses to Questions

**Comments to the Author**

1. If the authors have adequately addressed your comments raised in a previous round of review and you feel that this manuscript is now acceptable for publication, you may indicate that here to bypass the “Comments to the Author” section, enter your conflict of interest statement in the “Confidential to Editor” section, and submit your "Accept" recommendation.

Reviewer #1: (No Response)

Reviewer #2: (No Response)

2. Is the manuscript technically sound, and do the data support the conclusions?

Reviewer #1: Yes

Reviewer #2: No

3. Has the statistical analysis been performed appropriately and rigorously? 

Reviewer #1: Yes

Reviewer #2: Yes

4. Have the authors made all data underlying the findings in their manuscript fully available?

Reviewer #1: Yes

Reviewer #2: Yes

5. Is the manuscript presented in an intelligible fashion and written in standard English?

Reviewer #1: Yes

Reviewer #2: No

6. Review Comments to the Author

Reviewer #1: I thank the authors for the responses to the first review and the work undertaken to improve the readability of the manuscript. By positioning the interpretation on the strength of relationship as compared to the significance greatly improves the application of the outcomes. The remaining concerns are primarily related to clarifying the methodology.

Ln 52; "evaluated via rate of force development (RFD) of a force-time curve generated during..."

Ln 61; 'stronger'

Ln 134-156; The authors highlighted in the introduction that IMTP and ISqt have been used in the past but that this may be limited by upper body grip strength of compromised due to injury concerns, however each of these tests have also had the reliability and validity of the methods comprehensively reported. As such that authors need to include what is the CV%, and ICC for repeated testing using the MST device and also justify why the 100N pre tension has no impact. The citation provided (Ref #14) is on a very different device and procedure.

Ln 165; "shingle"?

Ln 197-198; The free fall formula is an unusual formula to be applied in this context because you don't know when the object began to fall, you are only aware of the total time in the air. Application of the impulse momentum theorem, and the Law of the Conservation of Energy. Please provide a reference for the formula and explain the constant.

Reviewer #2: General comments

A positive aspect of the study is that the authors used a rebound jump test which has not been frequently examined in the literature and measured a large sample of subjects. In general low to moderate correlations were found between IRFDs measured at different time points and VJ performance parameters. However, the authors conclude that “the early phase IRFD measured by ISLP could enable us to assess multiple aspects of leg extension strength characteristics, which will be useful to monitor the effect of strength training on an improvement of leg extension strength capabilities”. I do not believe that the data presented support such a strong statement. The authors should make changes in their discussion considering that IRFD measurements during a single leg press explain 3 to 13% of the variance in VJ performance.

In addition, the manuscript is not well written and is very confusing for the reader to understand the rationale of the authors and the procedures used for testing. The manuscript should be carefully checked for English language grammar and syntax.

Specific comments

Lines 127-131. This sentence is very complex and not well written.

Lines 164-167. I am sorry but I do not understand these sentence…

Line 165. Change shingle to single.

With the term CMJ most of the researchers refer to a jump with the hands placed on the waist. In the present study, the authors used a CMJ with an arm swing. This should be made clear even from the introduction and I suggest the use of CMJas as an abbreviation in order to be easier for the reader to understand exactly the type of the jump used.

Why did the subjects executed only two trials of CMJs? Are the authors confident that they recorded accurately the vertical jump ability of the subjects.

Line 199. Why the authors analyzed only the best jump and not the average of the two jumps. Wouldn’t this be more representative of subject’s ability?

Lines 278-281, 289, 294-295…. I am confused… At the methods section the authors write that “The IRFDs were obtained as rates of increases in force across 50 ms in the interval from the onset to 250 ms as shown in Fig 3 (IRFD0–50, IRFD50–100, IRFD100–150, IRFD150–200 and IRFD200–250)”. The same terminology is used at the tables. When I read IRFD100-150 I understand that the change in force from 100 ms to 150 ms is calculated. However, the authors in the discussion write for example, IRFDs 0-150ms. When I read this I understand that the change in force from the onset to first 150 ms is calculated. This is different from 100-150ms. What is correct? The authors should make this clear from the methods section and consistent in the abbreviations they use throughout the text.

In the discussion, the authors based on for the significant correlations they found between the measured IRFDs and vertical jump variables try to explain the reasons for these relationships. Though, significant correlations can be found when a large sample of subjects is measured as the authors did in the present study and they are commended for that. A significant correlation does not mean a large effect as well. In the present study, only small to moderate correlations were found (0.165 – 0.457) between the iRFDs measured at different time intervals and VJ performance variables. Therefore, IRFDs explain from only 2.7 up to 20.9% of the variance in VJ performance parameters. When the partial correlations are considered, the significant correlations range from 0.174 to 0.362 explaining only 3 to 13% of the variance in VJ performance. Therefore, I do not agree with the approach the authors follow in the discussion where the importance these type of testing is overemphasized while the results do not support that.

Based on my previous comment I do not agree with lines 372-379 and 385-387

Table 1. Change “Index” to “Reactive strength index”

7. PLOS authors have the option to publish the peer review history of their article (what does this mean?). If published, this will include your full peer review and any attached files.

Reviewer #1: **Yes: **Dale Wilson Chapman

Reviewer #2: No

---

## [Author Response · Author response to Decision Letter 1]

29 Apr 2023

Response to the comments

Manuscript ID# PONE-D-21-32499

The title of manuscript: 

Association of multi-phase rates of force development during an isometric leg press with vertical jump performances

Reviewer #1: I thank the authors for the responses to the first review and the work undertaken to improve the readability of the manuscript. By positioning the interpretation on the strength of relationship as compared to the significance greatly improves the application of the outcomes. The remaining concerns are primarily related to clarifying the methodology.

Response to Reviewer # 1

Ln 52; "evaluated via rate of force development (RFD) of a force-time curve generated during..."

[Response]

Yes, thank you for pointing it out. I corrected it. (L55)

Ln 61; 'stronger'

[Response]

Yes, thank you for pointing it out. I corrected it. (L64)

Ln 134-156; The authors highlighted in the introduction that IMTP and ISq have been used in the past but that this may be limited by upper body grip strength of compromised due to injury concerns, however each of these tests have also had the reliability and validity of the methods comprehensively reported. As such that authors need to include what is the CV%, and ICC for repeated testing using the MST device and also justify why the 100N pre-tension has no impact. The citation provided (Ref #14) is on a very different device and procedure.

[Response]

I stated CV% and ICC of IRFDs and PF in the statistical analysis and results (Ln 219-225, Ln 236-239, Table 3). Influence regarding pre-tension has been included in the Ln 229-234, Table 2 of the manuscript. We also reported the impact of pre-tension on IRFD using correlation coefficients and discussed the potential contribution of pre-tension to improving the ICC and CV% of IRFD (L314-328).

Ln 165; "shingle"?

[Response]

Yes, thank you for pointing it out. It was a simple spelling mistake. I corrected it. (L170)

Ln 197-198; The free fall formula is an unusual formula to be applied in this context because you don't know when the object began to fall, you are only aware of the total time in the air. Application of the impulse momentum theorem, and the Law of the Conservation of Energy. Please provide a reference for the formula and explain the constant.

[Response]

One of general method for calculating jump height in field tests is to use the total time in the air, and its reliability and validity have been demonstrated in previous research (García-López et al., 2005; Healy et al., 2016; Montalvo et al., 2021; Zushi et al., 2022). In addition, a very high correlation has been found between performance calculated from ground reaction forces during VJ and performances calculated from the total time in the air(Aragón, LF. 2000). There facts suggest that the difference in calculation methods is unlikely to have a significant effect on the results of this study. To demonstrate that we have controlled the factors that affect the calculation of jump height using the total time in the air. (L173-180) 

Response to Reviewer # 2

General comments

A positive aspect of the study is that the authors used a rebound jump test which has not been frequently examined in the literature and measured a large sample of subjects. In general low to moderate correlations were found between IRFDs measured at different time points and VJ performance parameters. However, the authors conclude that “the early phase IRFD measured by ISLP could enable us to assess multiple aspects of leg extension strength characteristics, which will be useful to monitor the effect of strength training on an improvement of leg extension strength capabilities”. I do not believe that the data presented support such a strong statement. The authors should make changes in their discussion considering that IRFD measurements during a single leg press explain 3 to 13% of the variance in VJ performance.

In addition, the manuscript is not well written and is very confusing for the reader to understand the rationale of the authors and the procedures used for testing. The manuscript should be carefully checked for English language grammar and syntax.

[Response]

Thank you for reviewing our study. I have revised the specific comments based on the review provided in the general comments.

Specific comments

Lines 127-131. This sentence is very complex and not well written. 

[Response]

I reduced the complexity of the sentence. Please confirm. (Lines 127-132)

Lines 164-167. I am sorry but I do not understand these sentence…

[Response]

I reduced the complexity of the sentence. Please confirm. (Lines 168-171)

Line 165. Change shingle to single. 

[Response]

Yes, thank you for pointing it out. It was a simple spelling mistake. We have revised. (Lines 170)

With the term CMJ most of the researchers refer to a jump with the hands placed on the waist. In the present study, the authors used a CMJ. This should be made clear even from the introduction and I suggest the use of CMJas as an abbreviation in order to be easier for the reader to understand exactly the type of the jump used.

Why did the subjects executed only two trials of CMJs? Are the authors confident that they recorded accurately the vertical jump ability of the subjects. Line 199. Why the authors analyzed only the best jump and not the average of the two jumps. Wouldn’t this be more representative of subject’s ability?

[Response]

I had changed CMJ to CMJAS, RJ to RJAS and, VJ to VJAS. we have described a protocol for the number of VJAS trials (Line 175). To measure many athletes within a limited schedule, we seated at least two successful VJAS trials. In addition, we conducted at least three familiarization trials while monitoring performance variables to ensure reliability and validity. We have revised the manuscript (Lines 172-173, Lines 175-180). Using the average of two trials may underestimate the participants' abilities, so we considered it necessary to analyze the "best jump" to evaluate the highest level of performances that participants could achieve at the test

Lines 278-281, 289, 294-295…. I am confused… At the methods section the authors write that “The IRFDs were obtained as rates of increases in force across 50 ms in the interval from the onset to 250 ms as shown in Fig 3 (IRFD0–50, IRFD50–100, IRFD100–150, IRFD150–200 and IRFD200–250)”. The same terminology is used at the tables. When I read IRFD100-150 I understand that the change in force from 100 ms to 150 ms is calculated. However, the authors in the discussion write for example, IRFDs 0-150ms. When I read this I understand that the change in force from the onset to first 150 ms is calculated. This is different from 100-150ms. What is correct? The authors should make this clear from the methods section and consistent in the abbreviations they use throughout the text.

[Response]

In the discussion, the term "IRFDs 0-150ms" refers to the IRFDs calculated for three separate time intervals: IRFD0-50, IRFD50-100, and IRFD100-150. As pointed out, We have standardized the wording throughout the entire text to either “IRFD0-50, IRFD50-100, and IRFD100-150” or expressions that correctly identify the IRFDss used in this study, such as “IRFDs were calculated from onset to ~ ms”, “IRFDs during onset to ~ ms ”, “IRFDs durig onset to ~ ms ” and so on (Lines 307-310, Lines 334-335, Line 348, Line 393, Line 409, Line 411, Lines 427-431). 

In the discussion, the authors based on for the significant correlations they found between the measured IRFDs and vertical jump variables try to explain the reasons for these relationships. Though, significant correlations can be found when a large sample of subjects is measured as the authors did in the present study and they are commended for that. A significant correlation does not mean a large effect as well. In the present study, only small to moderate correlations were found (0.165 – 0.457) between the iRFDs measured at different time intervals and VJ performance variables. Therefore, IRFDs explain from only 2.7 up to 20.9% of the variance in VJ performance parameters. When the partial correlations are considered, the significant correlations range from 0.174 to 0.362 explaining only 3 to 13% of the variance in VJ performance. Therefore, I do not agree with the approach the authors follow in the discussion where the importance these type of testing is overemphasized while the results do not support that. Based on my previous comment I do not agree with lines 372-379 and 385-387.

[Response]

We have revised the manuscript based on your review. Please confirm (Lines 414-418, Lines 424-426, Lines 427-437).

Table 1. Change “Index” to “Reactive strength index”

[Response]

I had changed “Index” to “Reactive strength index” or ”RSI”. Please confirm. (Table 1-5, Lines 210…)

---

## [Decision Letter · Decision Letter 2]

22 May 2023

PONE-D-21-32499R2Association of multi-phase rates of force development during an isometric leg press with vertical jump performancesPLOS ONE

Dear Dr. Zushi,

Thank you for submitting your manuscript to PLOS ONE. After careful consideration, we feel that it has merit but does not fully meet PLOS ONE’s publication criteria as it currently stands. Therefore, we invite you to submit a revised version of the manuscript that addresses the points raised during the review process.

You will see that both reviewers commend the revisions and feel the manuscript is better. They do, however, note some inconsistencies and lack of clarity at points, which would benefit the manuscript further. 

We look forward to receiving your revised manuscript.

Kind regards,

Theodoros M. Bampouras

Academic Editor

PLOS ONE

Journal Requirements:

Reviewers' comments:

Reviewer's Responses to Questions

**Comments to the Author**

1. If the authors have adequately addressed your comments raised in a previous round of review and you feel that this manuscript is now acceptable for publication, you may indicate that here to bypass the “Comments to the Author” section, enter your conflict of interest statement in the “Confidential to Editor” section, and submit your "Accept" recommendation.

Reviewer #1: (No Response)

Reviewer #2: All comments have been addressed

2. Is the manuscript technically sound, and do the data support the conclusions?

Reviewer #1: Yes

Reviewer #2: Yes

3. Has the statistical analysis been performed appropriately and rigorously? 

Reviewer #1: Yes

Reviewer #2: Yes

4. Have the authors made all data underlying the findings in their manuscript fully available?

Reviewer #1: Yes

Reviewer #2: Yes

5. Is the manuscript presented in an intelligible fashion and written in standard English?

Reviewer #1: Yes

Reviewer #2: No

6. Review Comments to the Author

Reviewer #1: I think the authors for the completed revisions, I have several further comments and suggestions.

Ln202-204; Can the authors please include a justification for why only the dominant leg GRF was used for comparison to the ISLP data which is described as using bipedal force production? Quite simply the strength of the relationships could have been influenced by this change in seeking to compare the dynamic force production capabilities of a single leg to the force generation capacity of both legs in a different position/set up.

Ln213; Please confirm that the data was checked for normal distribution prior to conducting the Pearson's correlation analysis

Ln309-311; While I agree with the statement from the authors, it needs to be clarified that the VJ data was from a single dominant leg compared to bipedal force production and as such has likely influenced the strength of the correlation coefficients.

Reviewer #2: I thank the authors for considering my suggestions. The manuscript has improved significantly, and now the discussion section is better based on the findings of the study.

Still, I believe that the whole manuscript should be checked very carefully for grammar and syntax issues and that there are several parts that are unclear to the reader (for example, lines 204-206, lines 220-222, lines 235-238, lines 315-318, lines 426-428).

Lines 222-224: what is the purpose of these sentence in the statistical analyses section?

Line 228: I cannot understand what this is.... In the methods it is written that a standard force value of 100N was used as a pretension. If it was a standard value for all the participants how a correlation would exist? Maybe I do not understand something, and I would like the authors to explain better this issue.

Table 1. First line, it should be 95% CI and not 95% IC.

7. PLOS authors have the option to publish the peer review history of their article (what does this mean?). If published, this will include your full peer review and any attached files.

Reviewer #1: **Yes: **Dale WILSON CHAPMAN

Reviewer #2: No

---

## [Author Response · Author response to Decision Letter 2]

6 Jul 2023

Response to the comments

Manuscript ID# PONE-D-21-32499

The title of manuscript: 

Association of multi-phase rates of force development during an isometric leg press with vertical jump performances

Reviewer #1: I think the authors for the completed revisions, I have several further comments and suggestions.

Response to Reviewer # 1

Ln202-204; Can the authors please include a justification for why only the dominant leg GRF was used for comparison to the ISLP data which is described as using bipedal force production? Quite simply the strength of the relationships could have been influenced by this change in seeking to compare the dynamic force production capabilities of a single leg to the force generation capacity of both legs in a different position/set up.

 [Response]

Taking into consideration the characteristics of single-leg and double-leg movements and intending to make participants’ measurements easier (line 134-136 and line 168-173). we employed the single leg press and bilateral VJ in this study. In the date processing for VJs, we used the dominant leg (line 207-208) that was used for isometric leg press measurement (line 136-137).

Ln213; Please confirm that the data was checked for normal distribution prior to conducting the Pearson's correlation analysis.'

[Response]

Several variables did not exhibit a normal distribution upon analyzing the normality (p > 0.05, line 224-225). Various sports athletes participated in the study can attributed to the results. Therefore, we have reanalyzed suitable statistical analyses (line 224-230). It is worth noting that the reanalysis did not yield significant changes in the results (line 233-290, Table1-5).

Ln309-311; While I agree with the statement from the authors, it needs to be clarified that the VJ data was from a single dominant leg compared to bipedal force production and as such has likely influenced the strength of the correlation coefficients. 

[Response]

If we can improve the homogeneity of the participants and make the positions and/or setups of VJ and ISLP more similar, there is a possibility that the relationship between VJ and rapid force production capabilities could become stronger. I have added the topic (line414-416).

 

Reviewer #2: I thank the authors for considering my suggestions. The manuscript has improved significantly, and now the discussion section is better based on the findings of the study

Response to Reviewer #2

Still, I believe that the whole manuscript should be checked very carefully for grammar and syntax issues and that there are several parts that are unclear to the reader (for example, lines 204-206, lines 220-222, lines 235-238, lines 315-318, lines 426-428). 

[Response]

We had the manuscript proofread by an English editing service and have checked the sentences clearer in terms of grammar and syntax.

Lines 222-224: what is the purpose of these sentence in the statistical analyses section?

[Response]

These sentences are the responses to comment form the other reviewer, and their purpose is to justify the reliability and validity of VJ measurements (line 219-221).

Line 228: I cannot understand what this is.... In the methods it is written that a standard force value of 100N was used as a pretension. If it was a standard value for all the participants how a correlation would exist? Maybe I do not understand something, and I would like the authors to explain better this issue.

[Response]

In this study, we instructed the participants to perform pre-tension approximately 100 N through visual estimation using the monitor (line149-151). This does not imply a requirement for the precise execution of a 100 N pre-tension. Because precise adjustments of muscle strength could potentially lead to neural inhibition and have a negative impact on explosive force generation. Therefore, considering the comment from the other reviewer, we investigated the influence of pre-tension that was not precisely 100N on the measurement values (line 238-240, Table 2, line 312-326).

Table 1. First line, it should be 95% CI and not 95% IC.

[Response]

Tank you for pointing it out. It was a simple spelling mistake. I corrected it (Table 1).

---

## [Editor Report · Decision Letter 3]

17 Jul 2023

PONE-D-21-32499R3Association of multi-phase rates of force development during an isometric leg press with vertical jump performancesPLOS ONE

Dear Dr. Zushi,

Thank you for submitting your manuscript to PLOS ONE. After careful consideration, we feel that it has merit but does not fully meet PLOS ONE’s publication criteria as it currently stands. Therefore, we invite you to submit a revised version of the manuscript that addresses the points raised during the review process.

The review should focus on two points:1. Answering the point made by Reviewer 1 with regards to the issues and potential limitations of comparing bilateral to unilateral tasks. Although an addition was made (Lines 414-416), the addition is not satisfactory as it does not really address this important issue. Please revisit this with a rationale for this selection (in Methods) and add any limitations this could cause (in the Limitations).2. I appreciate the revised manuscript benefited from proof-reading. There remain, however, several passages where language is still difficult to follow, which might result in the point you are conveying being unclear. I would therefore, suggest that the revision includes another proof-reading, to ensure clarity of language for all points made. 

We look forward to receiving your revised manuscript.

Kind regards,

Theodoros M. Bampouras

Academic Editor

PLOS ONE
---

## [Author Response · Author response to Decision Letter 3]

18 Sep 2023

Response to the comments

Manuscript ID# PONE-D-21-32499

The title of manuscript: 

Association of multi-phase rates of force development during an isometric leg press with vertical jump performances

The review should focus on two points:

1. Answering the point made by Reviewer 1 with regards to the issues and potential limitations of comparing bilateral to unilateral tasks. Although an addition was made (Lines 414-416), the addition is not satisfactory as it does not really address this important issue. Please revisit this with a rationale for this selection (in Methods) and add any limitations this could cause (in the Limitations).

[Response]

Thank you for pointing it out. In methods, I have reconsidered the reasons for using the double-leg vertical jump (VJ) and the single-leg leg press (line 134-139, 172-177). In addition to scientific rationale, these measurement designs were chosen because of considering how to enable diverse sports of participants to easily attain high performance in this measurement. Furthermore, I have suggested through the reference of a previous study that there are no critical issues of comparing bilateral to unilateral tasks(line 178-180). 

 However, there are some issues, I have added the limitations, including the comparison between bilateral and unilateral tasks (line 430-437).

2. I appreciate the revised manuscript benefited from proof-reading. There remain, however, several passages where language is still difficult to follow, which might result in the point you are conveying being unclear. I would therefore, suggest that the revision includes another proof-reading, to ensure clarity of language for all points made. 

[Response]

We had the manuscript proofread by an English editing service and have checked the sentences clearer in terms of grammar and syntax.

---

## [Editor Report · Decision Letter 4]

21 Sep 2023

Association of multi-phase rates of force development during an isometric leg press with vertical jump performances

PONE-D-21-32499R4

Dear Dr. Zushi,

We’re pleased to inform you that your manuscript has been judged scientifically suitable for publication and will be formally accepted for publication once it meets all outstanding technical requirements.

Kind regards,

Theodoros M. Bampouras

Academic Editor

PLOS ONE
---

## [Editor Report · Acceptance letter]

6 Nov 2023

PONE-D-21-32499R4 

Association of multi-phase rates of force development during an isometric leg press with vertical jump performances 

Dear Dr. Zushi:

I'm pleased to inform you that your manuscript has been deemed suitable for publication in PLOS ONE. Congratulations! Your manuscript is now with our production department. 

Kind regards, 

on behalf of

Dr. Theodoros M. Bampouras 

Academic Editor

PLOS ONE